# Study of Energy Loss Characteristics of a Shaft Tubular Pump Device Based on the Entropy Production Method

**DOI:** 10.3390/e25070995

**Published:** 2023-06-29

**Authors:** Dongtao Ji, Weigang Lu, Bo Xu, Lei Xu, Linguang Lu

**Affiliations:** College of Hydraulic Science and Engineering, Yangzhou University, Yangzhou 225009, China; dx120200089@yzu.edu.cn (D.J.); xubo@yzu.edu.cn (B.X.); leixu@yzu.edu.cn (L.X.); lglu@yzu.edu.cn (L.L.)

**Keywords:** entropy production method, energy loss, shaft tubular pump device, model test, numerical simulation

## Abstract

The unstable flow of a shaft tubular pump device (STPD) leads to energy loss, thereby reducing its efficiency. The aim of this study is to investigate the distribution pattern of energy loss in STPDs. This paper reveals that the two components with the highest proportion of energy loss are the impeller and the outlet passage. Furthermore, turbulent entropy production is the primary cause of energy loss. Due to the wall effect, the energy loss in the impeller mainly occurs near the hub and shroud. Additionally, the presence of a tip leakage vortex near the shroud further contributes to the energy loss in the region near the shroud. This results in the energy loss proportion exceeding 40% in the region with a volume fraction of 14% near the shroud. In the outlet passage, the energy loss mainly occurs in the front region, with a volume fraction of 30%, and the energy loss in this part accounts for more than 65%. Finally, this study reveals the locations of the vortex in the STPD under different flow-rate conditions, and when the distribution of energy loss is visualized, it is found that the energy loss occurs high in the vortex regions.

## 1. Introduction

Water resources are indispensable for the production and life of human beings; however, with the occurrence of floods and droughts caused by global climate variation, many new pumping stations have been built worldwide to meet the continuously increasing needs of water resource allocation [1,2]. In China, water resources are severely unevenly distributed in space and time, and to improve the present situation, the government has invested in the construction of the famous South-to-North Water Transfer Project, of which the east line project is composed of numerous large-scale pumping stations. These large-scale pumping stations provide huge amounts of flow and generally operate for more than 3000 h annually [3]. Based on such operating conditions, the pumping device used in these pumping stations must have excellent efficiency and stability. After being developed for decades, the shaft tubular pump device (STPD) has become the most widely used type of pump device in coastal areas of China for its excellent hydraulic performance, and up to now, the STPD has been adopted in about 100 pumping stations. In the past, the hydraulic performance and optimized design of the STPD have already been studied [4,5,6]. Xu et al. [7] optimized the design of the STPD by numerical simulation method and tested its performance. The results showed that the optimal efficiency of the device exceeded 83%. However, with the acceleration of abnormal climate variation induced by increasing carbon emissions, China has proposed a national strategy for the peaking of CO_2_ emissions and carbon neutrality, which requires that pump devices as energy consumers must operate more efficiently. Therefore, conducting a more in-depth study on the energy loss mechanism of STPDs is of great significance.

In the past, hydraulic loss in most studies was typically calculated based on the pressure drop across two sections [8]. This method can only provide the value of the hydraulic loss of a domain but cannot determine where the loss occurs, so it has obvious limitations. Currently, there are a number of research studies that show that it is possible to calculate the loss by using the entropy production method. The implementation of the method is conducted by integrating the entropy production rate over the flow domain. This method is initially applied in simple studies of laminar flow, turbulent flow [9,10] and boundary layer flow [11,12]. Based on this method, Schmandt et al. [13] obtained the hydraulic loss coefficients for various conduits and discovered that the majority of the loss is concentrated downstream of the conduit. Herwig et al. [14] completed a correlation table between the actual roughness and the equivalent roughness. This method not only provides the loss value but also allows observation of where the loss occurs, making it a useful tool for optimizing the design of hydraulic machinery [15,16] and investigating energy loss in complex flow [17,18,19]. Gong et al. [20] applied entropy production theory in the study of a large Francis turbine firstly, and the results showed that the entropy production method can effectively locate high-loss areas. Li et al. [21] compared the hydraulic loss obtained by using the entropy production and pressure drop methods. The results showed that when the entropy production in the wall region (EPW) is taken into account, the difference between the two methods is small. Zhang et al. [22] found that the energy loss in the runner and volute of the centrifugal pump accounts for 30% and 60%. Under all working conditions, the turbulent entropy production (TEP) and EPW are important components of loss, while the direct entropy production can be ignored. Lu et al. [23] investigated the relationship between flow pattern and loss. It was found that flow separation in the runner is the main reason for high energy loss under low-flow-rate conditions, and the distribution of energy loss in the draft is closely related to the vortex rope.

Although the entropy production method has been proven to be a reliable research method in the field of rotating machinery in many studies, it has not been applied in the analysis of energy loss characteristics of STPDs. In this paper, numerical simulations under multiple operation conditions are conducted by CFX software and the accuracy of the simulations is validated through a model test. The energy loss of the STPD and its components are obtained by using both the entropy production and pressure drop methods, and the distribution characteristics and variation laws under different flow conditions are explored. By combining the flow field with the distribution of entropy production rate, the relationship between the internal flow and energy loss in STPDs can be verified. The results can provide a theoretical basis for further optimization of STPDs.

This article consists of the following sections. Section 2 describes the research object and the numerical simulation method. Section 3 provides the detailed information related to the model test. In Section 4, two methods for evaluating energy loss are introduced. The detailed discussion and results about this study are presented in Section 5. Finally, the conclusions are proposed in Section 6.

## 2. Numerical Simulation

### 2.1. Settings and Turbulence Model

The STPD consists of four parts, and the three-dimensional model is shown in Figure 1. The basic parameters of the STPD are shown in Table 1. The dynamic–static interface is set as a frozen rotor in steady simulation and a transient rotor stator in the transient simulation. The common settings are shown in Table 2. In this study, the SST (shear stress transport) *k*-*ω* turbulence model proposed by Menter [24,25] is chosen for the simulation.

### 2.2. Grid Division and Scheme Selection

In this study, the parts of an STPD are divided into structured meshes. ICEM software is used for the mesh generation of two passages, and Turbo-Grid software is used for the grid division of the impeller and guide vanes. The grid diagram is shown in Figure 2.

Numerous studies have shown that the grid division of the simulation domains has an important influence on the results [26,27]. Therefore, in this study, the efficiency of STPDs is used as the control index for grid independence analysis. Five grid schemes are simulated in this study. Table 3 indicates that the simulation results are stable when the number of grid cells reaches 8.6 million, with an efficiency change of only 0.01%, which can be considered negligible.

The grid convergence index (GCI) proposed by Roache is a commonly applied method to verify the reliability of the mesh, and the validity of the GCI has been verified by many scholars [28,29,30]. The procedure of GCI calculation in this study is shown in Table 4. As shown in the table, the GCI is less than 2%, which indicates that the dispersion error of the simulation is small. Considering the grid independence and GCI, the 8.6 million grid scheme is used for numerical simulation.

The distribution of Yplus in the four components of the STPD is shown in Figure 3, and the average Yplus values are 29.33, 16.61, 23.37 and 24.87, respectively. According to the Yplus values in the numerical simulations in refs. [15,31], the mesh scheme in this study can be used.

## 3. Model Test

The high-precision test rig is located at the China Water Resources Beifang Co., Ltd. located in Tianjing, China, as shown in Figure 4. The measurement uncertainty of the V15712-HD1A1D7D electromagnetic flow meter is ±0.2%. The LDG-500s differential pressure transmitter is used to test the head, and the measurement uncertainty is ±0.1%. The measurement uncertainty of the JCZL2-500 torque and speed sensor is ±0.1%. Based on the measurement uncertainty of the instruments mentioned above, the uncertainty of the system can be determined as ±0.24%. This model test meets the requirements of IEC standards. This model test tested the hydraulic performance of the STPD at multiple blade angles. Table 5 shows the 10 times repeatability test results for the highest-efficiency point with blade angle of −2 degrees, from which the random uncertainty of 0.2% can be deduced; finally, the comprehensive uncertainty of this experiment is obtained as 0.5%.

## 4. Analysis Method

### 4.1. Traditional Pressure Drop Method

In order to evaluate the energy conversion performance of each domain in the STPD, the hydraulic loss can be calculated using total pressure drop between the inlet and outlet of each part. In the past, the calculations of hydraulic loss in most studies were based on this method [32,33]. The hydraulic loss Δ*h_s_* in the stationary domain (inlet and outlet passage, guide vanes) and Δ*h_r_* in the rotating domain (impeller) can be calculated by Equations (1) and (2), respectively. The above hydraulic loss can be converted into the energy loss by Equation (3).
(1)$$∆h_{s}=\frac{p_{1}-p_{2}}{\rho g}$$
(2)$$∆h_{r}=\frac{P}{\rho gQ}-\frac{p_{2}-p_{1}}{\rho g}$$
(3)$$P_{∆h}=\rho gQ∆h$$
where *p*_1_ and *p*_2_ represent the total pressure of inlet and outlet sections, Pa; and *P* is the input shaft power, kW.

### 4.2. Entropy Production Method

Near the wall region, the viscous force leads the kinetic and pressure energy to be converted into internal energy, thus leading to the increase in entropy. In the high Reynolds number region, the unstable flow causes an increase in entropy production. Therefore, the entropy production method can be used to calculate the energy loss in the STPD, and the equations involved in the entropy production method can be referred to Ref. [21]. The turbulence kinetic energy equations for a Newtonian fluid in Cartesian coordinates could be expressed as follows:(4)$$\frac{D}{D_{t}}\left(\frac{1}{2}u_{i}u_{i}\right)=u_{i}F_{x_{i}}+\frac{1}{\rho }\frac{\partial \left(m_{ji}u_{i}\right)}{\partial x_{j}}-\frac{1}{\rho }\frac{\partial \left(pu_{i}\right)}{\partial x_{j}}\delta_{ij}+\frac{p}{\rho }\frac{\partial u_{i}}{\partial x_{j}}\delta_{ij}-\frac{m_{ji}}{\rho }\frac{\partial u_{i}}{\partial x_{j}}$$
(5)$$m_{ji}=\mu \left(\frac{\partial u_{i}}{\partial x_{j}}+\frac{\partial u_{j}}{\partial x_{i}}\right)-\frac{2}{3}\mu \delta_{ij}\frac{\partial u_{k}}{\partial x_{k}}$$

Using the velocity components *u*_1_, *u*_2_ and *u*_3_, the Φ can be expanded to Equation (6):(6)$$\Phi =m_{ji}\frac{\partial u_{i}}{\partial x_{j}}$$

The incompressible fluid satisfies the following continuity equation:(7)$$\frac{\partial u_{1}}{\partial x_{1}}+\frac{\partial u_{2}}{\partial x_{2}}+\frac{\partial u_{3}}{\partial x_{3}}=0$$

Therefore, Equation (6) can be simplified to Equation (8).
(8)$$\Phi =2\mu \left[{\left(\frac{\partial u_{1}}{\partial x_{1}}\right)}^{2}+{\left(\frac{\partial u_{2}}{\partial x_{2}}\right)}^{2}+{\left(\frac{\partial u_{3}}{\partial x_{3}}\right)}^{2}\right]\;+\mu \left[{\left(\frac{\partial u_{2}}{\partial x_{1}}+\frac{\partial u_{1}}{\partial x_{2}}\right)}^{2}+{\left(\frac{\partial u_{3}}{\partial x_{1}}+\frac{\partial u_{1}}{\partial x_{3}}\right)}^{2}+{\left(\frac{\partial u_{2}}{\partial x_{3}}+\frac{\partial u_{3}}{\partial x_{2}}\right)}^{2}\right]$$

The local entropy production rate (LEPR) can be calculated by using Equation (9).
(9)$${\dot{S}}_{D}^{\prime \prime \prime }=\frac{\dot{Q}}{T}$$

In the turbulent flow, the LEPR includes the direct entropy production rate induced by time-averaged motion and turbulent entropy production rate (TEPR) due to turbulent dissipation.
(10)$${\dot{S}}_{D}^{\prime \prime \prime }={\dot{S}}_{\bar{D}}^{\prime \prime \prime }+{\dot{S}}_{D^{\prime }}^{\prime \prime \prime }$$

The direct entropy production rate can be obtained by using Equation (11).
(11)$${\dot{S}}_{\bar{D}}^{\prime \prime \prime }=\frac{2\mu }{T}\left[{\left(\frac{\partial u_{1}}{\partial x_{1}}\right)}^{2}+{\left(\frac{\partial u_{2}}{\partial x_{2}}\right)}^{2}+{\left(\frac{\partial u_{3}}{\partial x_{3}}\right)}^{2}\right]\;+\frac{\mu }{T}\left[{\left(\frac{\partial u_{2}}{\partial x_{1}}+\frac{\partial u_{1}}{\partial x_{2}}\right)}^{2}+{\left(\frac{\partial u_{3}}{\partial x_{1}}+\frac{\partial u_{1}}{\partial x_{3}}\right)}^{2}+{\left(\frac{\partial u_{2}}{\partial x_{3}}+\frac{\partial u_{3}}{\partial x_{2}}\right)}^{2}\right]$$

Mathieu and Scott proposed a method to obtain the TEPR in the *k*-*ω* model. The formula is as follows [34]:(12)$${\dot{S}}_{D^{\prime }}^{\prime \prime \prime }=\beta \frac{\rho \omega k}{T}$$

The local entropy production (LEP) includes direct entropy production and TEP:(13)$$P_{SD}{=P}_{S\bar{D}}+P_{SD^{\prime }}=\int_{V}{\dot{S}}_{\bar{D}}^{\prime \prime \prime }dV\cdot T+\int_{V}{\dot{S}}_{D^{\prime }}^{\prime \prime \prime }dV\cdot T$$

In this study, the EPW can be calculated by using Equation (14).
(14)$$P_{SW}=\int_{A}\frac{\bar{\tau }\cdot \bar{v}}{T}dA\cdot T$$

The total entropy production can be calculated by using Equation (15).
(15)$$P_{s}=P_{SD}+P_{SW}{=P}_{S\bar{D}}+P_{SD^{\prime }}+P_{SW}$$

## 5. Results and Analysis

### 5.1. Validation of Simulation and Comparison of Energy Analysis Methods

The test results showed that the variation pattern of hydraulic performance under different angles is basically the same, so the STPD with a blade angle of 0 degrees is selected for detailed analysis. The energy performance curves of the STPD obtained by the model test and numerical simulation are shown in Figure 5. At 0.8 *Q_d_*~1.2 *Q_d_*, the head and the shaft power curves of the STPD decrease with the increase in flow rate. The efficiency curve of the STPD increases with the increasing flow rate, reaching a maximum at 1.0 *Q_d_* and then decreasing at 1.0 *Q_d_*. The relative errors of head, shaft power and efficiency between the model test and numerical simulation are 2.49%, 2.69% and 0.62%, respectively. The maximum error of the shaft power and efficiency occurs at 0.8 *Q_d_*, and the relative errors are 4.95% and 1.80%, respectively. The relative error at low-flow-rate conditions is mainly due to the flow separation at the blades. The maximum error of the head occurs at 1.2 *Q_d_* with a relative error of 5.2%, which is caused by the low absolute value of the head at high-flow-rate conditions. Although there are still some uncertainties in numerical simulations that may cause slight deviations from experimental results, overall, the results are reliable.

As shown in Figure 6, although the energy loss obtained by the entropy production method is slightly smaller than that obtained by the pressure drop method at each condition, the two methods still exhibit good uniformity. The variation trend of energy loss obtained by the two methods is the same; the energy loss decreases first and then increases in the flow range of 0.8 *Q_d_*~1.2 *Q_d_*. The results obtained by the entropy production method are proved to be reliable according to the traditional pressure drop method; hence, this study can be conducted based on the entropy production method.

### 5.2. Analysis of Energy Loss in the STPD

As shown in Figure 7, the variation trend of energy loss in each component obtained by the two methods is consistent. In the inlet passage, the energy loss is very small and increases monotonically, which is due to the increasing flow velocity. In the remaining three parts, the energy loss decreases first and then increases, and the energy loss is minimal at the design flow condition. In the range of 0.8 *Q_d_*~1.1 *Q_d_*, the loss proportion of each component from the largest to the smallest is impeller, outlet passage, guide vanes and inlet passage; however, the loss proportion of the outlet passage at 1.2 *Q_d_* is more than the impeller. According to Figure 7a, in the flow range of 0.8 *Q_d_*~1.2 *Q_d_*, the loss in the inlet passage accounts for 3.0~8.2% of the loss in the STPD, the loss in the impeller accounts for 30.9~47.7%, the loss in the guide vanes accounts for 10.2~26.2% and the loss in the outlet passage accounts for 31.5~41.4%. As shown in Figure 7b, the percentage of energy loss in each component is basically consistent with the results obtained by the traditional pressure drop method. In addition, the proportions of different types of entropy production are shown in Figure 7b. It is obvious that the DEP in each component is small and can be neglected, while the TEP is the main contributor to energy loss in STPDs, which is consistent with the conclusions of previous research [31,35].

### 5.3. Analysis of TEP Distribution

According to the analysis in Section 5.2, the energy loss in STPDs is mainly concentrated in the impeller, guide vane and outlet passage, and the main source of energy loss is TEP. This section will further analyze the distribution of TEP in these three components. The impeller and the guide vanes are evenly divided into 10 subdomains along the axial and radial directions. The subdomains of the impeller along the axial direction are named IA1~IA10, the subdomains along the radial direction are named IR1~IR10, the subdomains of the guide vanes along the axial direction are named GA1~IA10 and the subdomains along the radial direction are named GR1~IR10, as shown in Figure 8.

Figure 9 shows the TEP distribution of each subdomain of the impeller. At 0.8 *Q_d_*~1.0 *Q_d_*, the TEP gradually decreases, while in the flow range of 1.0 *Q_d_*~1.2 *Q_d_*, the TEP increases, and the trend of variation of the energy loss is consistent with Figure 7. As shown in Figure 9a, it can be found that the TEP in the impeller firstly increases and then decreases along the axial direction at each condition. In IA3, the flow separation at the leading edge of the impeller leads to a surge of TEP. At 0.8 *Q_d_*, the TEP in IA3 is the largest, which is due to the decrease in the attack angle; the flow separation is more serious, as is the formation of backflow and vortex [36]. It can be found from Figure 9b that the TEP in the middle subdomains of the impeller is relatively stable, which indicates that the flow state in this region is stable. As it is influenced by the frictional resistance of the wall and the viscous resistance of the liquid, thus the flow in the axial direction near the wall is weakened and the loss increases [37]. Therefore, the TEP in IR1 and IR10 are clearly higher than that in the adjacent subdomains. Additionally, the presence of the tip leakage vortex near the shroud further promotes the TEP in IR10. This results in the TEP proportion exceeding 40% in IR10 with a volume fraction of 14%. Furthermore, the range of TLV increases as the flow velocity decreases [38], hence the TEP in IR10 is large at a small flow rate.

The guide vanes are positioned downstream of the impeller to reduce the circulation of the flow out of the impeller, and they enable the water to enter the outlet channel more smoothly. The distribution of TEP in the guide vanes is shown in Figure 10. According to Figure 10a, it can be found that at 0.9 *Q_d_*~1.2 *Q_d_*, the LEP in subdomains decreases along the axial direction, which indicates that the guide vanes reduce the circulation velocity of the flow out of the impeller, and finally the flow in the subdomains at the exit is stable and the TEP is lower. At 0.8 *Q_d_*, there is a vortex at the back of the guide vanes near the outlet section, therefore the TEP in GA10 increases. According to Figure 10b, along the radial direction, the TEP in GR1 and GR10 is clearly higher than that in the adjacent subdomains due to the wall effect. At 0.9 *Q_d_*~1.1 *Q_d_*, the TEP in the guide vanes is stable, which indicates that the flow is stable. At 0.8 *Q_d_* and 1.2 *Q_d_*, the distribution of TEP has no clear regularity due to the large deviation from the design flow rate, which also indicates the internal flow of guide vanes is chaotic.

Considering that the outlet passage is a leafless region, the distribution characteristics of TEP will be discussed only along the axial direction. According to Figure 11, the outlet passage is divided into 10 subdomains and named OA1~OA10 along the axial direction.

The TEP in each subdomain of the outlet passage is shown in Figure 12. At 0.9 *Q_d_*~1.1 *Q_d_*, the distribution characteristics of TEP along the axial direction are consistent. The TEP increases first, with the maximum in the OA3 and OA4, and then decreases gradually. The volume of OA1~OA4 accounts for 30%; however, the TEP accounts for more than 65%. In the rear part, the energy loss is very small because the velocity and circulation have decreased due to the completion of the flow diffusion. At 0.8 *Q_d_*, the TEP value in the OA1 is the largest, which is due to the effect of the velocity circulation and guide vanes’ wake vortex (GWV). At 1.2 *Q_d_*, due to the larger flow velocity, the TEP values in OA2~OA10 are larger than those at other flow-rate conditions.

### 5.4. Visualization Analysis

Three typical operation conditions are selected for visualization and analysis. Figure 13 and Figure 14 show the distribution of TEPR and the flow characteristics in the impeller at the typical span of 0.03, 0.5 and 0.97. The location of the high TEPR region can be verified by the flow structure, and the TEPR is higher in the region of poor flow characteristics. At all conditions, there is an impeller wake vortex (IWV) at the trailing edge of the blade resulting in a small high TEPR region. Near the hub (span = 0.03), there is a high TEPR region at the leading edge of the impeller due to the vortex at the leading edge of impeller (ILV) at 0.8 *Q_d_*. The ILV is the flow separation caused by the large attack angle at a small flow rate. There is also a high TEPR region in the impeller passage, which is caused by the hub vortex of the impeller (IHV). At 1.0 *Q_d_* and 1.2 *Q_d_*, the attack angle decreases and the ILV disappears, hence the TEPR is significantly lower. In the middle passage (span = 0.5), the flow moves along the blade airfoil, hence the TEPR is obviously lower than that near the hub and shroud. Near the shroud region (span = 0.97), the TLV leads to the high TEPR region, and as the flow rate increases; the head and the pressure difference between the pressure side (PS) and suction side (SS) decrease, which leads to a reduction of the area of the TLV and high TEPR region.

Figure 15 and Figure 16 show the distribution of TEPR and the flow characteristics in the guide vanes at a typical span of 0.03, 0.5 and 0.97. Due to the presence of GWV at three operating conditions, there is a small high TEPR region at the trailing edge of the guide vanes. In addition, there is the guide vanes hub vortex (GHV) near the hub, which results in high TEPR at corresponding locations. As the flow rate increases, the GHV first decreases and then increases, and the high TEPR region shows the same variation. The matching relationship between the flow direction and the guide vanes placement angle is shown in Figure 17. At 0.8 *Q_d_*, because of the misfit of flow direction and the guide vanes placement angle, there is a guide vanes separation vortex (GSV) in the SS side near the outlet, and the TEPR is high in this region. At 1.0 *Q_d_*, the flow in the guide vanes is stable without flow separation, hence the TEPR is low. At 1.2 *Q_d_*, the location of GSV shifts to the PS of the guide vanes near the inlet and results in a high TEPR region.

In order to further investigate the distribution of the TEPR in the impeller and guide vanes, six monitoring sections are set up along the axial direction, as shown in Figure 18. Figure 19 shows the distribution of TEPR and flow characteristics in the monitoring sections of the impeller; from left to right, they are S1, S2 and S3. The distribution of TEPR and flow characteristics can verify each other, and the TEPR is high in the region with disordered flow. At each of the flow-rate conditions, the high TEPR regions are mainly distributed on the blade surfaces and near the hub and shroud regions due to the influence of IHV and TLV. As the IHV gradually increases along the axial direction, the area of the high TEPR region near the hub also increases.

Figure 20 shows the distribution of TEPR and flow characteristics in the monitoring sections of the guide vanes. In Figure 20, from left to right, they are S4, S5 and S6. The distribution of TEPR and flow characteristics can verify each other, and the TEPR is high in the region with disordered flow. At 0.8 *Q_d_*, the presence of GHV near the hub leads to a small high TEPR region. Along the axial direction, GSV appears in the SS of the guide vanes and gradually increases, hence the area of the high TEPR region also increases. When the flow rate is 1.0 *Q_d_* and the TEPR is small in each section of the guide vanes, there is only a small high TEPR region that appears near the hub caused by GHV and gradually increases with the axial direction. At 1.2 *Q_d_*, in each monitoring section, there are high TEPR regions near the hub and the PS of the guide vanes due to the influence of GSV and GHV. Along the axial direction, the high TEPR region gradually decreases because of the decrease in the GSV range.

Figure 21 shows the distribution of TERP in the horizontal and vertical sections of the outlet passage under each operation condition. According to Figure 21, it is found that the TEPR in the outlet passage has basically symmetrical distribution and is sensitive to the change of flow rate. The high TEPR region decreases and then increases with the increase in flow rate, which is because the energy loss in the outlet passage is influenced by both the circulation velocity and the flow velocity. The high TEPR region is completely concentrated in the front part of the outlet passage, which further validates the distribution law of energy loss in Figure 12. Six monitoring sections are set up along the axial direction; the location and numbers are shown in Figure 22.

Figure 23 shows the distribution of TEPR in the monitoring sections of the outflow passage. The common characteristics can be found that the high TEPR region in the sections decreases gradually along the axial direction. In S7~S9, the distribution of high TEPR is clearly influenced by the guide vanes, with six high TEPR regions appearing in each section, which is consistent with the number of guide vanes. The flow out of the guide vanes still has circulation, so the flow moves toward the outer wall by centrifugal force; therefore, there is a high TEPR region near the outer wall surface of each section, and the high TEPR region near the outer wall surface increases as the flow rate increases. In S11~S12, the flow has been adjusted by the outlet passage to complete the diffusion and kinetic energy recovery, so the flow is stable and therefore the TEPR is quite low, which is consistent with the law in Figure 12. Compared with Figure 20, it is found that the distribution characteristics of TEPR in S7 are basically the same as those in S6, but the intensity of TEPR is reduced; this indicates that the flow in the outlet passage is still influenced by GSV, GWV and GHV, and the influence gradually decreases along the axial direction.

## 6. Conclusions

In this study, the flow field and energy loss of an STPD are simulated, and the accuracy of the numerical simulation is verified by a model test. The energy loss characteristics at each working condition are analyzed based on the entropy production and pressure drop methods. The main achievements of this study are as follows:The energy loss in each part of the STPD is analyzed, and the results show that the prediction of the energy loss by the above two methods is consistent. The TEP is the primary cause of energy loss, while the EPW is the secondary source. TEP accounts for 65.40~77.88% of the energy loss in the STPD, whereas EPW accounts for 21.54~33.63%.At 0.8 *Q_d_*~1.1 *Q_d_*, the component with the largest energy loss in an STPD is the impeller. Due to the wall effect and TLV, the energy loss in the region near the impeller shroud accounts for more than 40% of the total loss in the impeller. Therefore, it is necessary to consider the influence of the tip clearance of the impeller to reduce losses.At 1.2 *Q_d_*, the component with the largest energy loss in an STPD is the outlet passage. The energy loss in the outlet passage is mainly concentrated in the entrance part due to the wake flow of the guide vanes. Therefore, the adoption of appropriate measures at the entrance part can be considered to eliminate the influence of the wake flow and reduce loss.The visualization of the TEPR of the main components of the STPD clearly identify the vortex regions, which are accurately verified by the flow field structure. The main vortex structures in the impeller are IWV, IHV and TLV, while the main vortex structures in the guide vanes are GWV, GHV and GSV. It is clearly found that the TEPR is significantly higher in the vortex region. The distribution of TEPR in the outlet passage is clearly influenced by the guide vanes, and the influence gradually decreases along the axial direction.

## Figures and Tables

**Figure 1 entropy-25-00995-f001:**
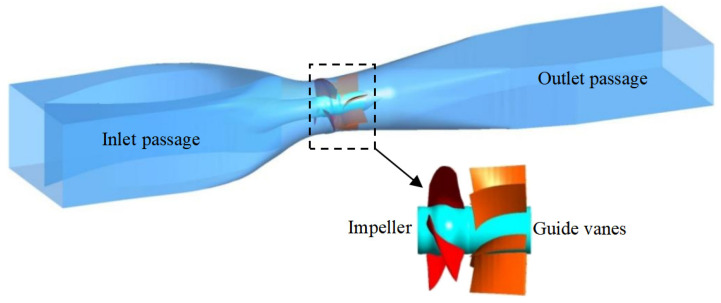
3D model of an STPD.

**Figure 2 entropy-25-00995-f002:**
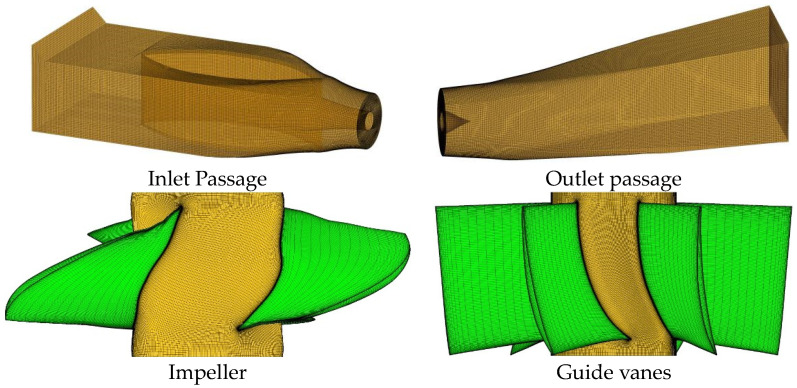
Grid diagram of an STPD.

**Figure 3 entropy-25-00995-f003:**
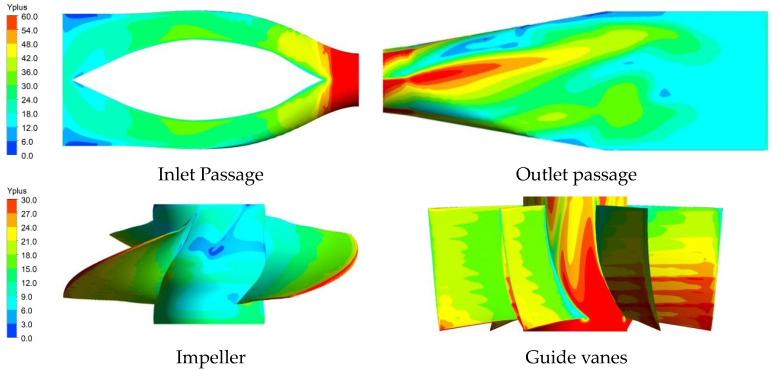
Distribution of Yplus.

**Figure 4 entropy-25-00995-f004:**
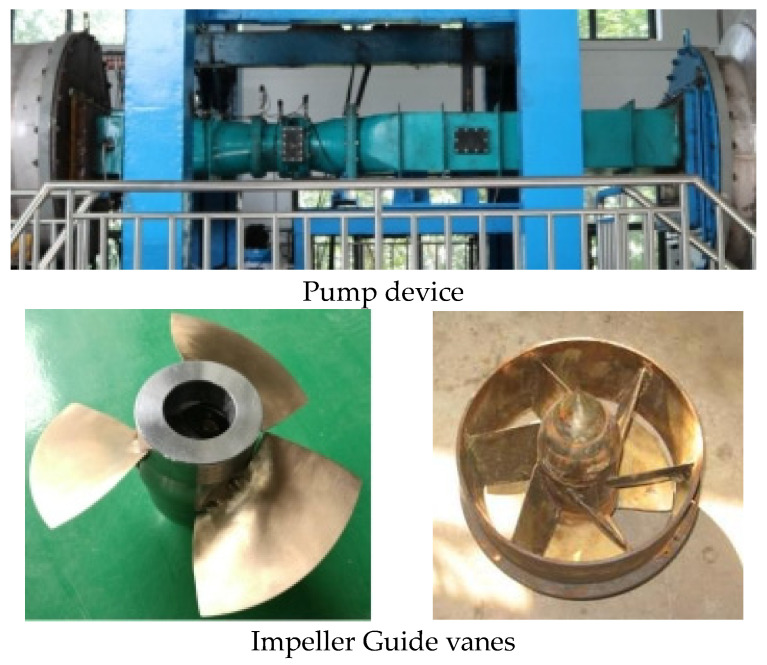
Physical diagrams of test.

**Figure 5 entropy-25-00995-f005:**
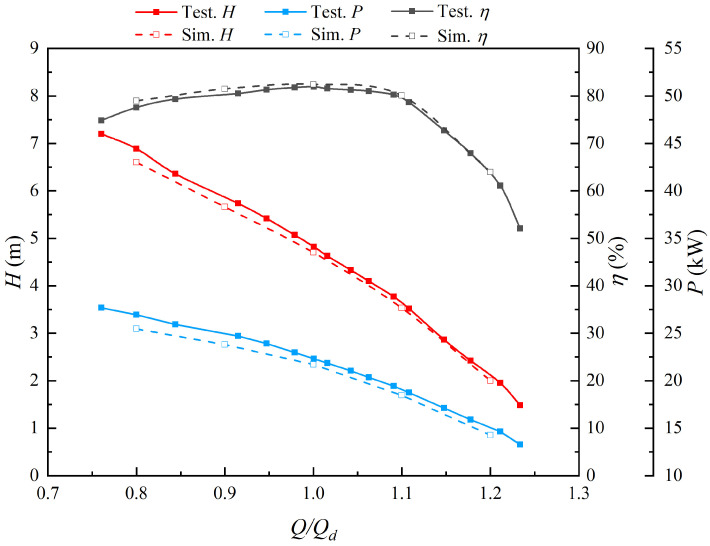
Comparison between simulation results and model test results.

**Figure 6 entropy-25-00995-f006:**
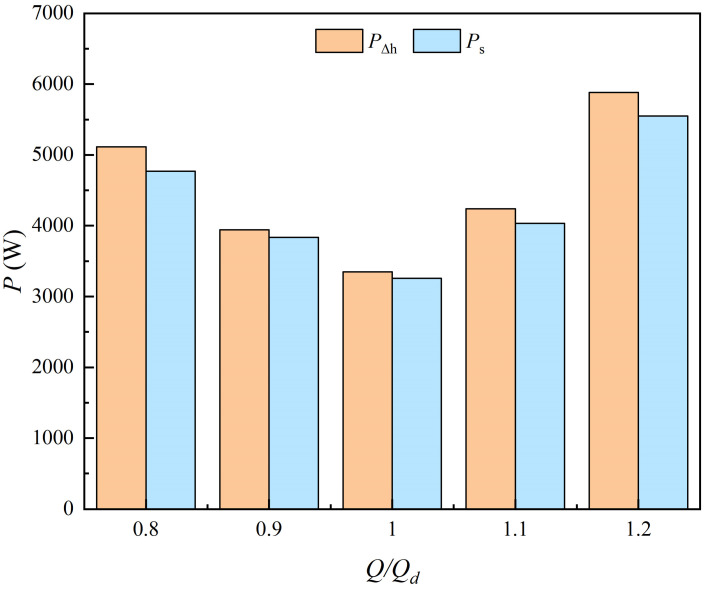
Comparison of the energy loss obtained by the two methods.

**Figure 7 entropy-25-00995-f007:**
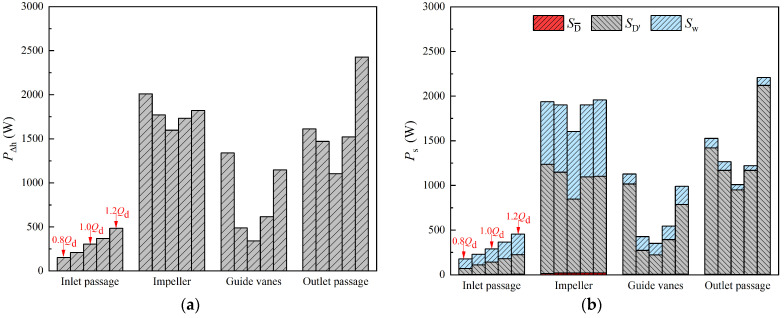
Energy loss in each part of an STPD at different flow-rate conditions. (**a**) Traditional pressure drop method; and (**b**) entropy production method.

**Figure 8 entropy-25-00995-f008:**
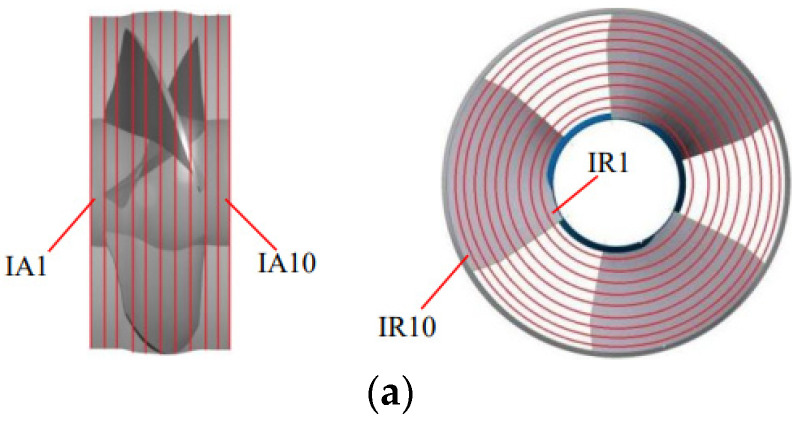
The division diagram. (**a**) The subdomains of the impeller; and (**b**) the subdomains of the guide vanes.

**Figure 9 entropy-25-00995-f009:**
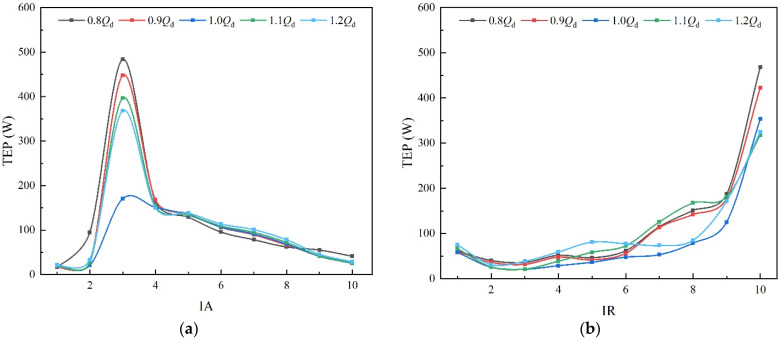
The distribution of TEP in the impeller: (**a**) axial direction; and (**b**) radial direction.

**Figure 10 entropy-25-00995-f010:**
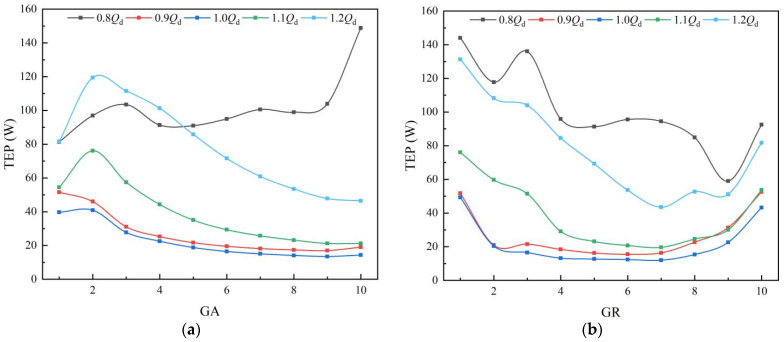
The distribution of TEP in guide vanes: (**a**) axial direction; and (**b**) radial direction.

**Figure 11 entropy-25-00995-f011:**
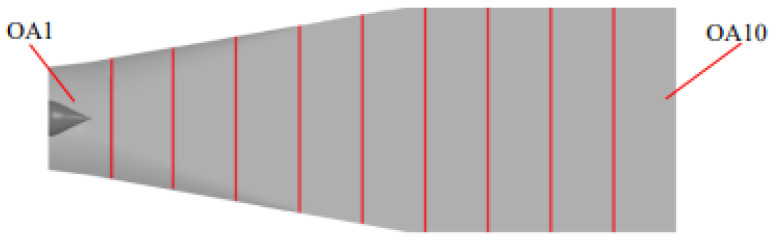
The division diagram of the outlet passage.

**Figure 12 entropy-25-00995-f012:**
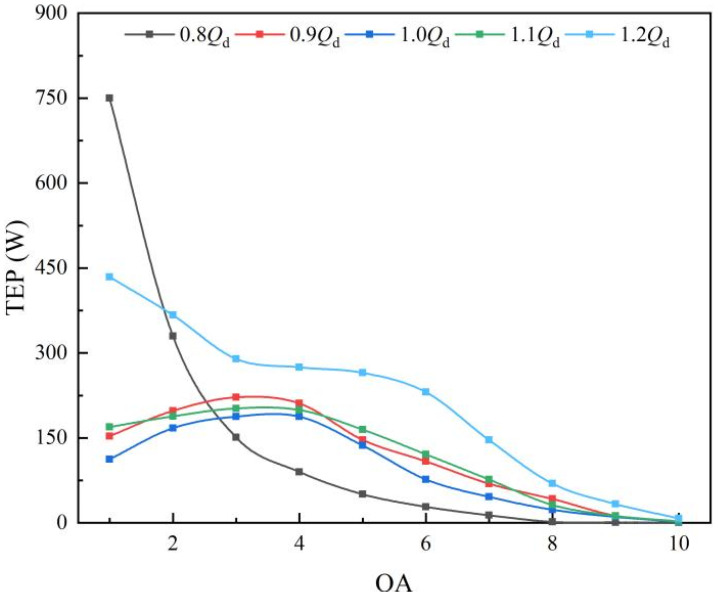
The distribution of TEP in the outlet passage.

**Figure 13 entropy-25-00995-f013:**
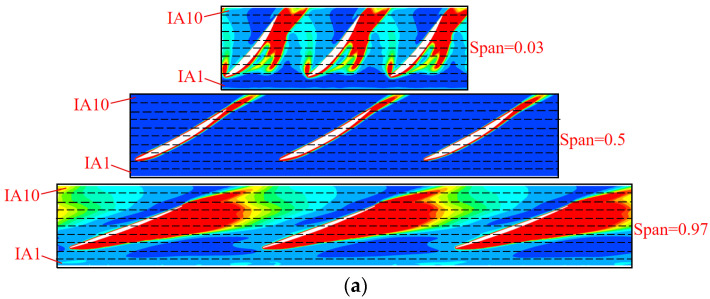
The distribution of TEPR in the impeller: (**a**) 0.8 *Q_d_*; (**b**) 1.0 *Q_d_*; and (**c**) 1.2 *Q_d_*.

**Figure 14 entropy-25-00995-f014:**
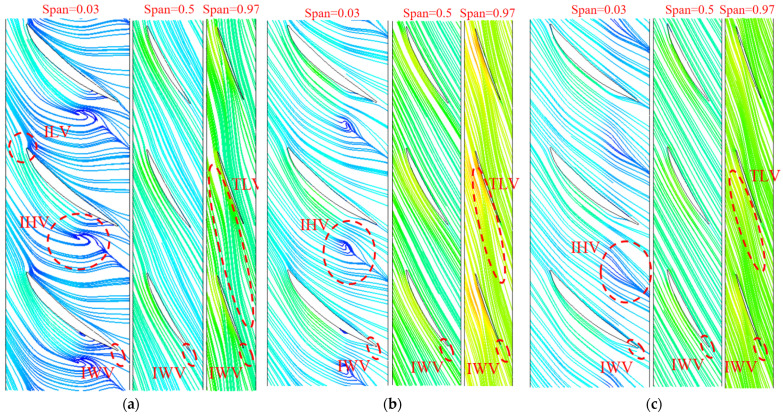
The flow characteristics in the impeller: (**a**) 0.8 *Q_d_*; (**b**) 1.0 *Q_d_*; and (**c**) 1.2 *Q_d_*.

**Figure 15 entropy-25-00995-f015:**
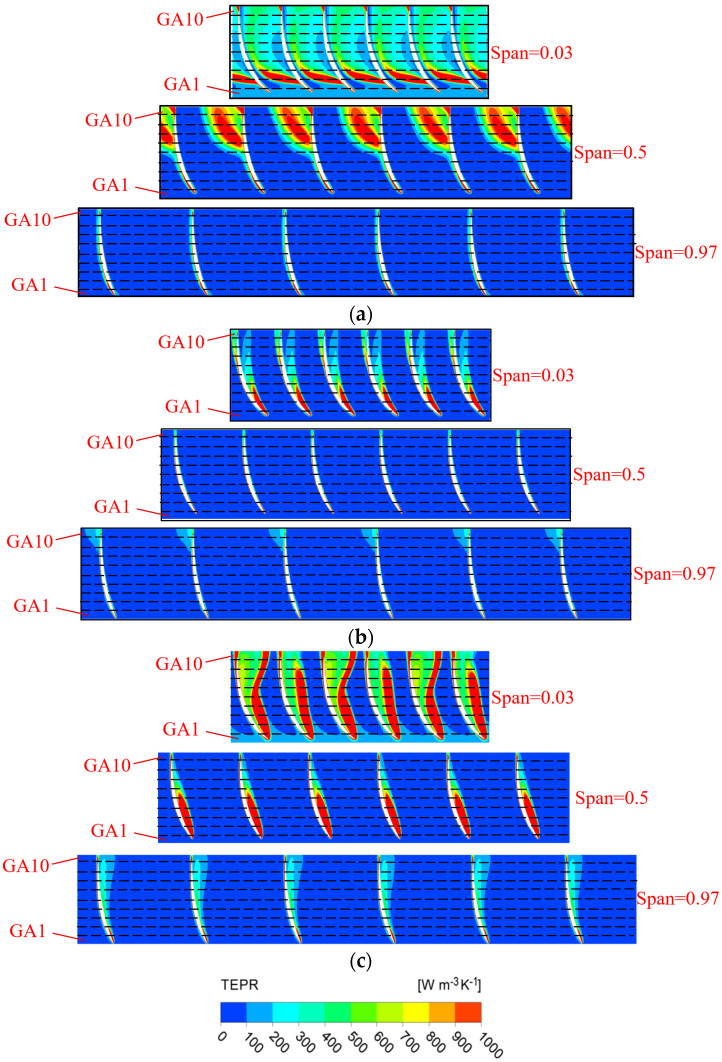
The distribution of TEPR in the guide vanes: (**a**) 0.8 *Q_d_*; (**b**) 1.0 *Q_d_*; and (**c**) 1.2 *Q_d_*.

**Figure 16 entropy-25-00995-f016:**
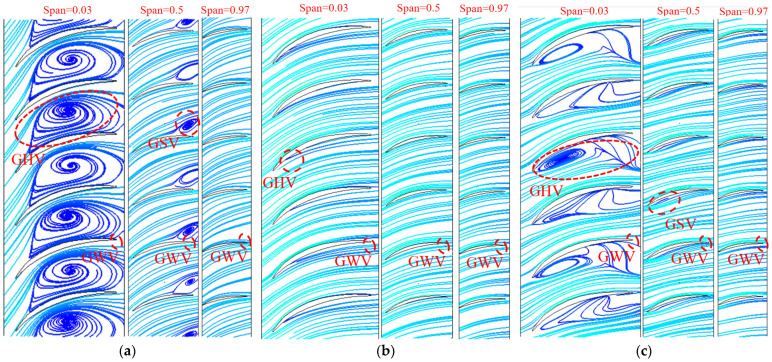
The flow characteristics in the guide vanes: (**a**) 0.8 *Q_d_*; (**b**) 1.0 *Q_d_*; and (**c**) 1.2 *Q_d_*.

**Figure 17 entropy-25-00995-f017:**
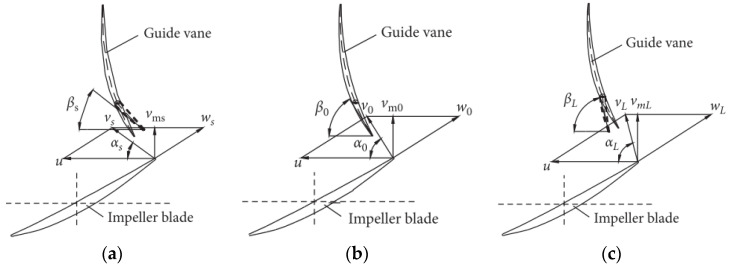
Matching relationship between the flow direction and the guide vanes placement angle: (**a**) 0.8 *Q_d_*; (**b**) 1.0 *Q_d_*; and (**c**) 1.2 *Q_d_*.

**Figure 18 entropy-25-00995-f018:**
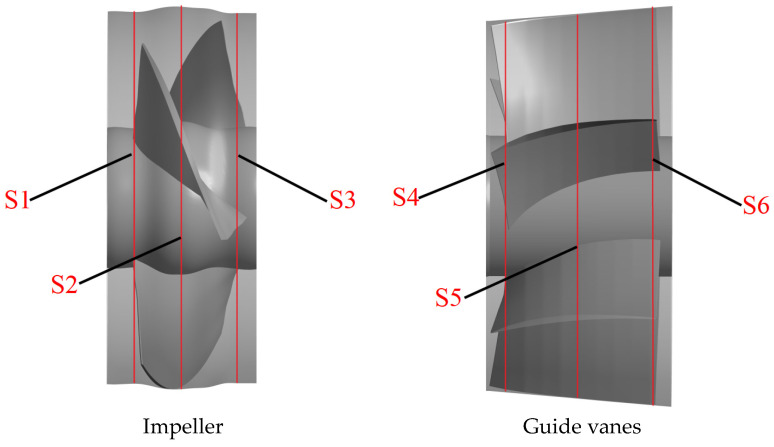
The location of monitor sections of the impeller and guide vanes.

**Figure 19 entropy-25-00995-f019:**
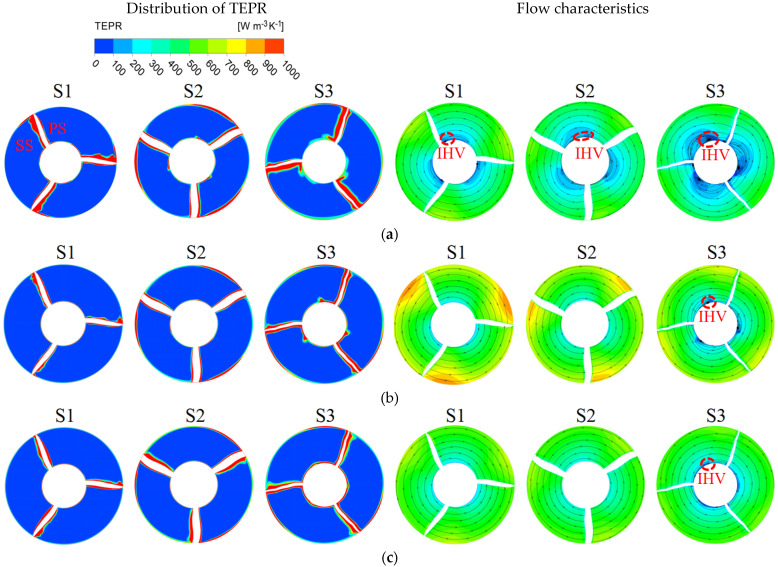
The distribution of TEPR and flow characteristics in the monitoring sections of the impeller: (**a**) 0.8 *Q_d_*; (**b**) 1.0 *Q_d_*; and (**c**) 1.2 *Q_d_*.

**Figure 20 entropy-25-00995-f020:**
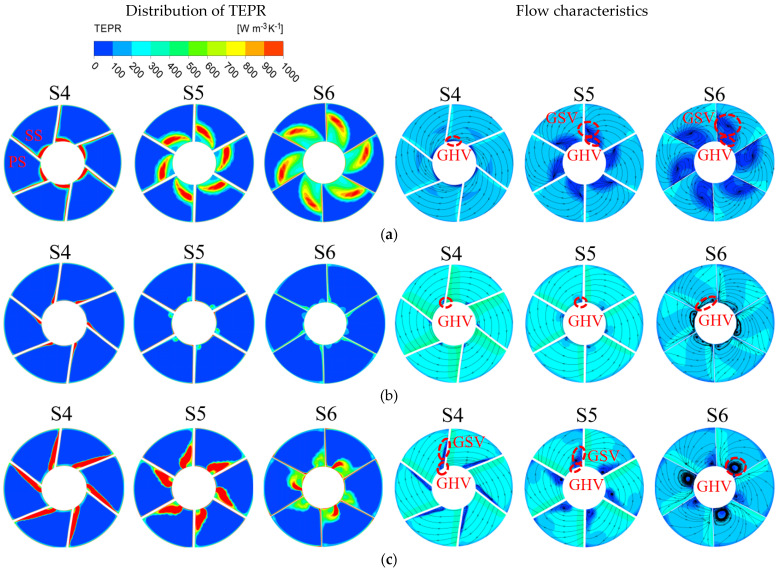
The distribution of TEPR and flow characteristics in the monitoring sections of the guide vanes: (**a**) 0.8 *Q_d_*; (**b**) 1.0 *Q_d_*; and (**c**) 1.2 *Q_d_*.

**Figure 21 entropy-25-00995-f021:**
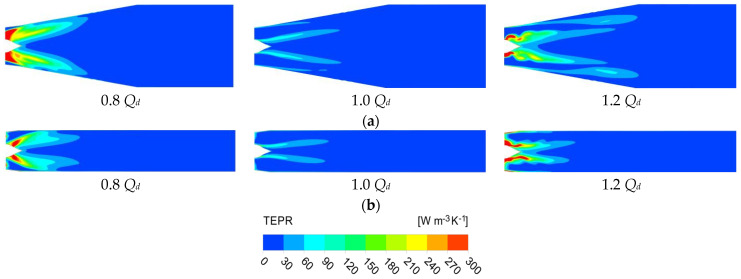
The distribution of TERP in outlet passage: (**a**) horizontal section; and (**b**) vertical section.

**Figure 22 entropy-25-00995-f022:**
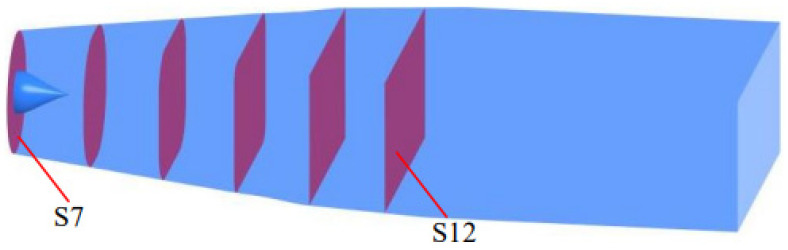
The location of monitoring sections of the outlet passage.

**Figure 23 entropy-25-00995-f023:**
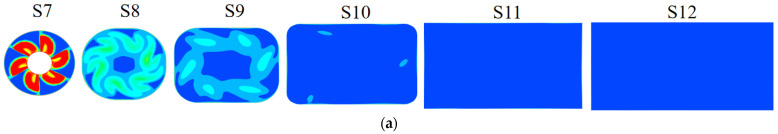
The distribution of TEPR in the monitoring sections of the outlet passage: (**a**) 0.8 *Q_d_*; (**b**) 1.0 *Q_d_*; and (**c**) 1.2 *Q_d_*.

**Table 1 entropy-25-00995-t001:** Parameters of an STPD.

Parameters	Values
Diameter of pump	300 mm
Impeller blade number	3
Guide vanes blade number	6
Tip clearance	0.2
Rotational speed	1450 r/min

**Table 2 entropy-25-00995-t002:** Common settings.

Parameters	Values
Inlet	Total pressure (1 atm)
Outlet	Mass flow rate
Wall	No-slip
Convergence accuracy	1 × 10^−4^
Time step	3.45 × 10^−4^ (s)
Total time	0.828 (s)

**Table 3 entropy-25-00995-t003:** Grid independence analysis.

Scheme	Number of Grid/10^6^	Efficiency/%
1	1.7	81.87
2	3.9	81.64
3	7.1	81.49
4	8.6	81.40
5	8.8	81.39

**Table 4 entropy-25-00995-t004:** The calculation of GCI.

Parameters	Values
*N*_1_, *N*_2_, *N*_3_	8.6 M, 3.9 M, 1.7 M
*r*_21_, *r*_32_	1.302, 1.319
*Φ*_1_, *Φ*_2_, *Φ*_3_	81.40%, 81.64%, 81.87%
*p*	2.28
$\Phi_{ext}^{21}$ *,* $\;\Phi_{ext}^{32}$	1.22, 1.21
$e_{a}^{21}$, $\;e_{a}^{32}$	0.29%, 0.17%
*GC1* _21_	0.44%
*GCI* _32_	0.24%

**Table 5 entropy-25-00995-t005:** Results of repeatability tests.

Number	1	2	3	4	5	6	7	8	9	10
Efficiency/%	82.12	82.18	82.04	82.26	82.09	82.15	82.27	82.20	82.19	82.22

## Data Availability

The data can be obtained by contacting the authors.

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
