# Peer review of "Study of Energy Loss Characteristics of a Shaft Tubular Pump Device Based on the Entropy Production Method"

_entropy, 2023, doi:10.3390/e25070995_

Round 1

Reviewer 1 Report

At the end of the Introduction, a paragraph should be added referring to the structure/organization of the manuscript.

Minor misprints:

end of line 273: a impeller should be an impeller

line 325: since the decreasing  --> because of the decreasing

line 369: better is occurs mainly

line 376: the text  should be deleted 

There are onlya few  minor misprints.

Author Response

Dear Editor and Reviewers,

Thank you for your affirmation of our work. And your valuable comments play a very important role in our paper. We are honored that you can help us point out these problems, because these problems in the article are all critical to the quality of the paper. At the same time, these suggestions are also of great benefit to our follow-up work.

Response to Reviewer 1 Comments:

At the end of the Introduction, a paragraph should be added referring to the structure or organization of the manuscript.

Response: Thanks for reviewer's comment. We have added a description of the manuscript structure at the end of the Introduction in the revised manuscript. The detailed modifications are shown in lines 80-84.

Minor misprints:

end of line 273: a impeller should be an impeller

Response: Thanks to the reviewer for pointing out some minor errors in our paper. We have corrected the error pointed out by the reviewer. Please see in the line 305.

line 325: since the decreasing → because of the decreasing

Response: Thanks to the reviewer for pointing out some minor errors in our paper. We have corrected the error pointed out by the reviewer. Please see in the line 367

line 369: better is occurs mainly

Response: Thanks to the reviewer for pointing out some minor errors in our paper. We have corrected in the revised manuscript.

line 376: the text should be deleted 

Response: Thanks to the reviewer for pointing out some minor errors in our paper. We have deleted this sentence in the revised manuscript and checked the manuscript again.

Reviewer 2 Report

Review

The research article titled “Study of energy loss characteristics of shaft tubular pump device based on entropy production method” gives an insight into the loss generation of a shaft tubular pump device (STPD). The STPD is divided into four sections, inlet passage, impeller, guide vanes and outlet passage, and the loss generation is analysed using the entropy production method. A comparison between the pressure drop method and the entropy production method reveals a satisfactory agreement between the two approaches.

Regrettably, the energy losses were not extensively localised and quantified; instead, it focuses on areas with a high proportion of losses. The assertion that losses occur predominantly in the outlet passage may be slightly misleading, as these losses mainly stem from the wake of the guide vanes. Expanding the section on guide vanes would also encompass wake losses, which primarily manifest in the front part of the outlet section.

Furthermore, it is worth noting that Section 4.2 on the entropy production method closely resembles the work of Li et al. (citation 20) without appropriate acknowledgment.

A critical aspect pertains to the consideration of Yplus, which with "averaged values" of 16.61 to 29.33 is stated to be sufficiently small. Firstly, the consideration should not rely on averaged values alone, and secondly, the citation Kan et al. (citation 31) provided as a justification for the adequacy of these values does not align with the statement. In the referenced text, a Yplus value of less than 1.5 is achieved in almost every cell within the boundary layer zone. It is therefore questionable whether the investigations with the present work are at all suitable, especially because the entropy production in the wall region and the wall effect are considered.

All in all, the paper has a good and logical structure. However, the reviewer suggests enhancing the precision of certain descriptions and justifying the requirement for Yplus.

Additional remarks:

Add the reference: Gong R Z, Wang H J, Chen L X, et al. Application of entropy production theory to hydro-turbine hydraulic analysis. Sci China Tech Sci, 2013, 56: 1636-1643, doi: 10.1007/s11431-013-5229-y

There are many abbreviations hidden in the text, so it would be better to make an abbreviation list

L. 102 number of grid “cells”

L. 119 Error on full size or measured value?

L. 123 “.“ This

L. 124 multiple blade angles are mentioned ... were different impellers examined? No adjustment possibility is visible.

L. 145 D“t“, in eqution (8) the last term should be

L. 173 Figure 5 is too small

LL. 192 – 207 Describe in more detail what the percentages refer to and reduce them to the essentials. The diagram says enough.

Figure 9, 10, 12 Absolute plot of losses interesting, but better related to input power

L. 232 A short description of the wall effect is missing.

L. 238 Introduce with guide vanes section

The entire section 5.4 is difficult to understand. Partly invisible vortices (L.274 IWV, 289 GWV) are listed or vortices with detachment are interchanged (Figure 16. GHV (a) SS, (c)PS). Suggestion: Draw a sketch like Lampard (Investigation of endwall flows and losses in axial turbines. Part I. Formation of endwall flows and losses ,January 2009, Journal of Theoretical and Applied Mechanics 47(2):321-34)

Figure 13 The scale at span 0.03, 0.5 and 0.97 is not constant. Plot the same axial length and set mark IA1-IA10.

Figure 18 & Figure 19 The monitor sections do not fit. Figure 19 what is the color in streamlines?

The scale in Figure 21 and 23 is misleading. Is 300 W sufficient for the first monitoring section?

L. 376 What is this???

Author Response

Dear Editor and Reviewers,

Thank you for your affirmation of our work. And your valuable comments play a very important role in our paper. We are honored that you can help us point out these problems, because these problems in the article are all critical to the quality of the paper. At the same time, these suggestions are also of great benefit to our follow-up work.

Response to Reviewer 2 Comments:

The research article titled “Study of energy loss characteristics of shaft tubular pump device based on entropy production method” gives an insight into the loss generation of a shaft tubular pump device (STPD). The STPD is divided into four sections, inlet passage, impeller, guide vanes and outlet passage, and the loss generation is analysed using the entropy production method. A comparison between the pressure drop method and the entropy production method reveals a satisfactory agreement between the two approaches.

Regrettably, the energy losses were not extensively localised and quantified; instead, it focuses on areas with a high proportion of losses. The assertion that losses occur predominantly in the outlet passage may be slightly misleading, as these losses mainly stem from the wake of the guide vanes. Expanding the section on guide vanes would also encompass wake losses, which primarily manifest in the front part of the outlet section.

Response: Thanks for reviewer's comment. In the practical application of the axial flow pump device, the guide vanes and the outlet flow passage are two separated parts. The guide vanes is made of metal while the outlet flow passage is constructed of concrete, and the interface is the exit section of the guide vanes. In practical applications, the size of the guide vanes is only enlarged according to the size of the impeller diameter in an equal scale, while the relative position of the outlet section cannot be changed. Although the loss in the entrance part of the outlet passage is significantly influenced by the wake of guide vanes, however, due to the size of the guide vane section cannot be expanded, hence the energy loss occurs in the outlet passage should be considered as the outlet passage energy loss.

Furthermore, it is worth noting that Section 4.2 on the entropy production method closely resembles the work of Li et al. (citation 20) without appropriate acknowledgment.

Response: Thanks for reviewer's comment.We have added an acknowledgement in the revised manuscript. Pleases see in the acknowledge.

A critical aspect pertains to the consideration of Yplus, which with "averaged values" of 16.61 to 29.33 is stated to be sufficiently small. Firstly, the consideration should not rely on averaged values alone, and secondly, the citation Kan et al. (citation 31) provided as a justification for the adequacy of these values does not align with the statement. In the referenced text, a Yplus value of less than 1.5 is achieved in almost every cell within the boundary layer zone. It is therefore questionable whether the investigations with the present work are at all suitable, especially because the entropy production in the wall region and the wall effect are considered.

All in all, the paper has a good and logical structure. However, the reviewer suggests enhancing the precision of certain descriptions and justifying the requirement for Yplus.

Response: Thanks for reviewer's comment. Regarding the range of Y+, we refer to some energy loss studies based on the entropy production method, and the turbulence model used in these studies is also the SST k-ω model. According to previous research results, we used similar Y+, so the results of this study can be considered reliable. Please see in the lines 122-123.  

Add the reference: Gong R Z, Wang H J, Chen L X, et al. Application of entropy production theory to hydro-turbine hydraulic analysis. Sci China Tech Sci, 2013, 56: 1636-1643, doi: 10.1007/s11431-013-5229-y

Response: Thanks to the reviewers for providing the references, we think this article is very helpful.We have cited this reference in the revised version. The detailed modifications are shown in the lines 58-60.

There are many abbreviations hidden in the text, so it would be better to make an abbreviation list

Response: Thanks for reviewer's comment. We have added a list of abbreviations in the revised manuscript.

L.102 number of grid “cells”

Response: Thanks to the reviewer for pointing out the minor error in our paper. We have corrected in the revised manuscript. Please see in the line 109.

L.119 Error on full size or measured value?

Response: Thanks for reviewer's comment. The error in the text refers to the measurement uncertainty of the experimental instrument. We have made corrections in the revised manuscript. Please see in the lines 127-131.

L.123 “.“ This

Response: Thanks to the reviewer for pointing out some minor errors in our paper. We have corrected in the revised manuscript. Please see in the line 132.

L.124 multiple blade angles are mentioned ... were different impellers examined? No adjustment possibility is visible.

Response: Thanks for reviewer's comment. We tested the hydraulic performance of the shaft tubular pump device under multiple blade angles, and the results showed that the variation pattern of hydraulic performance under different angles is basically the same, so the shaft tubular pump device with blade angle of 0 degree was selected for detailed analysis. We have clarified the blade angle in the revised version. Please see in the lines.178-180.

L.145 D“t“, in equation (8) the last term should be

Response: Thanks to the reviewer for pointing out some minor errors in our paper. We have corrected in the revised manuscript. Please see in the equation (5).

L.173 Figure 5 is too small

Response: Thanks for reviewer's comment.We have enlarged the size of Figure 5 in the revised manuscript.

LL. 192 – 207 Describe in more detail what the percentages refer to and reduce them to the essentials. The diagram says enough.

Response: Thanks to the reviewer's comment on the writing, and we have rewritten this paragraph in the revised manuscript.The detailed modifications are shown in the lines 211-219.

L.232 A short description of the wall effect is missing.

Response: Thanks for reviewer's comment. We have described the wall effect in the revised  manuscript. The detailed modifications are shown in the lines 259-262.

L.238 Introduce with guide vanes section

Response: Thanks for reviewer's comment. We have introduced the guide vanes in the revised manuscript. The detailed modifications are shown in the lines 268-270.

The entire section 5.4 is difficult to understand. Partly invisible vortices (L.274 IWV, 289 GWV) are listed or vortices with detachment are interchanged (Figure 16. GHV (a) SS, (c)PS). Suggestion: Draw a sketch like Lampard (Investigation of endwall flows and losses in axial turbines. Part I. Formation of endwall flows and losses ,January 2009, Journal of Theoretical and Applied Mechanics 47(2):321-342).

Figure 13 The scale at span 0.03, 0.5 and 0.97 is not constant. Plot the same axial length and set mark IA1-IA10.

Response: Thanks for reviewer's comment.We have modified Figures 13~16 to help the reader understand. The detailed modifications are shown in the Figures 13~16.

Figure 18 & Figure 19 The monitor sections do not fit. Figure 19 what is the color in streamlines?

Response: Thanks for reviewer's comment.We have modified the image and added the name of each section in the figure for better presentation. The color of the streamline is black. The detailed modifications are shown in the Figures 18-20.

The scale in Figure 21 and 23 is misleading. Is 300 W sufficient for the first monitoring section?

Response: Thanks for reviewer's comment.We are sorry that we ignored the possible misleading effect of the size, and we have made corrections in the revised manuscript. The detailed modifications are shown in the Figure 23.

L.376 What is this???

Response: Thanks for reviewer's comment. We have deleted this sentence in the revised manuscript and checked our manuscript again

Reviewer 3 Report

The article analyzes the problem of energy loss in a real shaft tubular pump device. The  method of analyse of entropy production was used. An important feature of the article is the combination of model research and experiment. The obtained results refer to a specific, tested object, so they are not fully universal. The scientific novelty of the research is limited, the research is technical in nature. Nevertheless, it is an example of comprehensive research and is worth publishing. However, authors should consider taking into account the following considerations:
1. Provide separate nomenclature for all abbreviations, variables, and parameters, including their units. Without this, the equations are not unambiguous.
2. Equation model (1,2,3) - only a reference to a literature, either a detailed description of the model should be used.
3. Figure 2 looks nice but conveys virtually no information about the grid.
4. Lines 119-121 - currently we don't use error but measurement uncertainty.
5. Chapter 4.1 - requires a broader description or only a reference to references. In this form, there is too much information for specialists, and too little information for others.
6. Figure 7a is cut...
7. Legends for visualization - recommended improvement of unit notation; additionally, the visualization material is very extensive, it should be limited to representative elements.
8. Conclusions are too concise and obvious. Maybe more cognitive and application information can be extracted from such extensive research?

Author Response

Dear Editor and Reviewers,

Thank you for your affirmation of our work. And your valuable comments play a very important role in our paper. We are honored that you can help us point out these problems, because these problems in the article are all critical to the quality of the paper. At the same time, these suggestions are also of great benefit to our follow-up work.

Response to Reviewer 3 Comments:

The article analyzes the problem of energy loss in a real shaft tubular pump device. The  method of analyse of entropy production was used. An important feature of the article is the combination of model research and experiment. The obtained results refer to a specific, tested object, so they are not fully universal. The scientific novelty of the research is limited, the research is technical in nature. Nevertheless, it is an example of comprehensive research and is worth publishing. However, authors should consider taking into account the following considerations:

  1. Provide separate nomenclature for all abbreviations, variables, and parameters, including their units. Without this, the equations are not unambiguous.

Response: Thanks for reviewer's comment. We have added a list of nomenclature and abbreviations in the revised manuscript.

  1. Equation model (1,2,3) - only a reference to a literature, either a detailed description of the model should be used.

Response: Thanks for reviewer's comment. We have cited the article of the proposer of this turbulence model as a reference in the revised manuscript. Please see in the lines 90-91.

  1. Figure 2 looks nice but conveys virtually no information about the grid.

Response: Thanks for reviewer's comment. Figure 2 is used to show that the meshes of each part are structured meshes, and the colors of the meshes have been modified for better presentation. Please see in the Figure 2.

  1. Lines 119-121 - currently we don't use error but measurement uncertainty.

Response: Thanks to the reviewer for pointing out our mistake. We have corrected it in the revised manuscript. Please see in the lines 127-131.

  1. Chapter 4.1 - requires a broader description or only a reference to references. In this form, there is too much information for specialists, and too little information for others.

Response: Thanks for reviewer's comment. We have described the pressure drop method in more detail in the revised manuscript. Please see in the lines 145-150.

  1. Figure 7a is cut...

Response: Thanks for reviewer's comment. We are sorry that the picture was not shown completely due to our negligence, and we have corrected it in the revised version.Please see in Figure 7a.

  1. Legends for visualization - recommended improvement of unit notation; additionally, the visualization material is very extensive, it should be limited to representative elements.

Response: Thanks for reviewer's comment. We have improved the picture in the revised manuscript. We provide detailed visualization material and the main purpose is to better demonstrate the distribution pattern of energy loss in each component.

  1. Conclusions are too concise and obvious. Maybe more cognitive and application information can be extracted from such extensive research?

Response: Thanks for reviewer's comment. We have modified the conclusions. The detailed modifications are shown in Conclusions section.
